# Peer review of "Biodegradable Mulching Films Based on Polycaprolactone and Its Porous Structure Construction"

_polymers, 2022, doi:10.3390/polym14245340_

Round 1

Reviewer 1 Report (Previous Reviewer 1)

The authors propose a new method for obtaining a biodegradable mulching film based on polycoprolactone. The results obtained by the authors do not convince that the proposed films are biodegradable. Within 2 months, only 50% film destruction is achieved. Is it possible to compare with other films already in use?

In addition, there are several comments.

Page 4: Rate cannot be expressed as a percentage.

Page 6: For a better understanding of the confirmation of the process of PL film degradation, it is necessary to present the IR spectrum of the original film.

Page 7: Due to what processes does the interaction of the film with water make the surface flatter?

Author Response

Dear reviewer:

Thank you for your decision and constructive comments on my manuscript. We have carefully considered the suggestion of Reviewer and make some changes. We have tried our best to improve and made some changes in the manuscript.

The yellow part that has been revised according to your comments. Revision notes, point-to-point, are given in the attachement.
best regards!
Jialei Liu

Reviewer 2 Report (New Reviewer)

This article is devoted to the production and study of films based on polycaprolactone. The article is interesting and has scientific novelty. The main goals and objectives are not in doubt. There are some important points that should be improved:

1. In the introduction, you can expand the description by adding an overview of other materials based on lactones. Please cite: 10.17516/1998-2836-0266, 10.1080/23312009.2018.1443689, 10.17516/1998-2836-2016-9-3-345-352.

2. Part of the data is a dataset. Please add a comparison with already published data. More comparative description would be desirable.

3. It is necessary to interconnect all methods of analysis performed by the authors. Now it looks disjointed.

4. It is desirable to add more physico-chemical methods for the analysis of the obtained products and/or experimental data.

5. Conclusion needs to be expanded.

6. Please cite: 10.3390/foods10112571.

In general, the article is interesting, but it needs to be expanded.

Author Response

Dear reviewer:

Thank you for your decision and constructive comments on my manuscript. We have carefully considered the suggestion of Reviewer and make some changes. We have tried our best to improve and made some changes in the manuscript.

The yellow part that has been revised according to your comments. Revision notes, point-to-point, are given in the attachement.
best regards!
Jialei Liu

Round 2

Reviewer 1 Report (Previous Reviewer 1)

The authors have tried to answer all the comments of the reviewer. Thank you.

Reviewer 2 Report (New Reviewer)

Accepted

This manuscript is a resubmission of an earlier submission. The following is a list of the peer review reports and author responses from that submission.

Round 1

Reviewer 1 Report

The article presents interesting results but their interpretation should be improved for better understanding and readability.

Page 1 Introduction. What means 1.425 mt? And what means 300 million mu?

Page 3. Subsection 3.1 Check if the link to the picture is correct. Fig. 2 are IR spectra. Due to what processes does a significant weight loss of the film occur at 4020C?

Page 4 Subsection 3.2 The authors present data on the IR spectra of the films obtained under different conditions. In the method of obtaining films, the method is indicated only with the use of trifluoroacetic acid. How did the authors obtain films in neutral and alkaline media considering that polycaprolactone is insoluble in water?

Page 5. Could the authors provide high magnification SEM images to confirm the nano-size of the pores and the presence of the pores themselves?

What can be associated with a uniform surface in Fig. 3 a4?

Page 6. Figures 4 and 5 can be combined. The authors argue that the lowest rate of film decomposition is recorded in an acidic environment. However, according to this Fig. 4 and 5, the lowest decomposition rate was obtained in a neutral environment.

Page 7, second paragraph: The role of acid or alkali is not entirely clear, since the decomposition of the films was carried out using dilute solutions.

Subsection 3.5: You should not start a new subsection with a table. For a better understanding of the data in Table 1, it is necessary to describe the detailed procedures of the experiments.

Conclusion. Check the text. There are repetitions of sentences (paragraphs (1) and (2)). The formulation of the conclusions does not correspond to the results presented in the main text.

Check the main text again. There are errors in references to certain figures

Author Response

Dear Reviewer

   Thanks for your positive suggestion about our manuscript. It really can help improve the quality of our paper. We have revised our manuscript according to your suggestion in details.

Best regards!

                                                             Jialei Liu

Reviewer 2 Report

The manuscript entitled "Preparation and characterization of porous biodegradable mulching films based on polycaprolactone" deals with the preparation and characterization of a porous PCL film.

In my view, the subject of the proposed work is not suitable for publication on Polymers. In fact, in the manuscript the simple preparation of a PCL film through TIPS is reported, without significant updates as compared to what already widely reported in the literature. The obtained film was then characterized through basic techniques. Once again, without adding new remarkable knowledge to the state of the art on PCL main properties. Additionally, the proposed preparation method of PCL film appears not immediately suitable for an industrial massive application, such as mulching films production. Finally, the English language of the whole manuscript should be carefully revised, aiming at improving its readability and avoiding misunderstandings (as an example, in the Abstract the Authors stated that "...Infrared spectroscopy (FI-IR) was used to break the chemical bonds during the degradation of different solutions...").

Author Response

Dear Reviewer

     Thanks for your positive suggestion about our manuscript. It really help us to improve our paper. We have revised our manuscript according to your suggestion. Please check it .

Best regards!

Jialei Liu

Reviewer 3 Report

Overall, the topic is interesting as it is related to bioplastics and sustainability, but there are many obvious issues as described below.

1. Introduction is not well written. Need to review biodegradable plastics for mulch film applications, e.g. PBAT, PHA, PBS, PLA, etc. Also need to  review any PCL related mulch film articles. Compare PCL with other biodegradable plastics and why they are interesting/superior and suitable for mulch film applications. Also why porous are interesting. 

2. Materials and methods are not well written. Many details are missing, e.g. sources, suppliers, molecular properties. For example, information of PE and PBAT and how to prepare their mulch film were not described. Sources and purity of many chemicals were not present. The information on degradation and sample preparation is not clear, e.g. the temperature, concentration of NaOH and HCl, the soil conditions (e.g. PH, temperature) which could all influence the degradation results. Information on measuring WVTR was not described. FTIR info, etc.

3. Results and discussion. Suggest to compare pure bulk PCL samples with treated PCL porous films to see how TFA influences PCL. What are the differences in crystallinity, thermal stability and Mw after treatment. TGA, DSC and GPC are required.

4. Results and discussion. The reasons that why porous structure of the film is important for the mulching films are not clear and well explained.

5. Results and discussion. Need to compare the pure PCL film (not porous) with porous PCL film on the temperature retention, WVTR, weeds restraining performance. And give more comprehensive explanations why porous structures are critical. Also how different porous size and porosity can influence the performance.

6. Moderate English changes required. Some confusing descriptions or arguments. Some obvious errors. E.g. 

“Infrared spectroscopy (FI-IR) was used to break the chemical bonds during the degradation of different solutions. FI-IR was used to study the breaking of the chemical bonds of the film during its degradation in different solutions. The mass loss method was used to characterize the degradation of the film material, and the degradation rate of PCL nano-porous films under different degradation environments.”

Fourier-transform infrared spectroscopy (FTIR?) How can IR be used to "BREAK" the chemical bonds? So confusing!!!

7. The conclusion is neither concise nor well organized. The reviewer would suggest the authors to rewrite the conclusion part.

8. There are some other minor issues.

Considering all of these, I would suggest to reconsider the manuscript after a major revision.

Author Response

(The authors gave the same response as above.)

Round 2

Reviewer 1 Report

The authors took into account all the comments of the reviewer. This increased the readability of the article.

Author Response

Dear reviewer

     Thanks for your nice comments on our article.

                             Best regards!

                                      Jialei Liu

Reviewer 2 Report

Despite the revisions made by the Authors, I'm still convinced (for the same reasons already stated during the first stage of revision, which were not fully adressed by the Authors) that the submitted manuscript is not suitable for publication on Polymers.  

Author Response

Dear reviewer:

Thank you for your positive comments on our article. according to your suggestions, we have corrected several mistakes in our previous draft. based on your comments, we have made extensive revisions to our previous draft.

  1. In my view, the subject of the proposed work is not suitable for publication on Polymers. In fact, in the manuscript the simple preparation of a PCL film through TIPS is reported, without significant updates as compared to what already widely reported in the literature.

TIPS is really not a new method for the processing of soluble polymers. But it is the first time for preparation of functional fully biodegradable mulch film. It is a new application in special areas. The design of porous structure on the surface enhances the heat preservation and moisture preservation function of this kind of mulch film. TIPS and tape casting process are not the conventional preparation methods of traditional PE mulch film, but they provide a new way for the development of functional fully biodegradable mulch film, especially the development and evaluation of new functional mulching film.

  1. The obtained film was then characterized through basic techniques.

Although the characterization of obtained film is a conventional method through basic techniques, the characterization content has included the requirements of the use process on the degradable mulch film, including structural characteristics, thermal stability, water retention, thermal insulation, degradability, etc. These indicators can support the application demand of such films in agriculture.

  1. Once again, without adding new remarkable knowledge to the state of the art on PCL main properties. Additionally, the proposed preparation method of PCL film appears not immediately suitable for an industrial massive application, such as mulching films production.

It is true that the plastic blow molding method is mainly used for the preparation of mulching film. However, with the replacement of PE by degradable plastics, the function of degradable mulching film will be expanded more in the future, and the traditional method of mulching film preparation will also be changed.

  1. Finally, the English language of the whole manuscript should be carefully revised, aiming at improving its readability and avoiding misunderstandings (as an example, in the Abstract the Authors stated that "...Infrared spectroscopy (FI-IR) was used to break the chemical bonds during the degradation of different solutions...").

The language of this manuscript has been checked and polished by native English speaker.

Round 3

Reviewer 2 Report

In my view, the submitted work is not suitable for publication on Polymers